# GC Insights: Space sector careers resources in the UK need a greater diversity of roles

Martin O. Archer[1], Cara L. Waters[1], Shafiat Dewan[1], Simon Foster[1], and Antonio Portas[2]

[1]Department of Physics, Imperial College London, London, UK
[2]NUSTEM, Northumbria University, Newcastle, UK

**Correspondence:** Martin O. Archer
(m.archer10@imperial.ac.uk)

**Abstract.** Educational research highlights that improved careers education is needed to increase participation in STEM. Current UK careers resources concerning the space sector, however, are found to perhaps not best reflect the diversity of roles present and may in fact perpetuate misconceptions about the usefulness of science. We, therefore, compile a more diverse set of space-related jobs, which will be used in the development of a new space careers resource.

## 1 Introduction

Educational research shows participation issues across Science Technology Engineering and Mathematics (STEM) are not due to school students' disinterest, but whether students see these fields and their potential career opportunities as for "people like me" (Archer and DeWitt, 2017, and references therein). These perceptions form early and remain relatively stable with age, which has led to recommendations for increased provision and quality of careers education/engagement at both primary and secondary levels (Archer et al., 2013; Holman, 2014; Davenport et al., 2020). Careers education provision in the UK specifically, however, is still not universal (despite mandates being in place) and that which is provided can often be patterned by societal inequities, unfortunately leaving some students' aspirations "dampened" (Abrahams, 2016; Archer and Moote, 2016; Moote and Archer, 2018a,b). It is therefore fair to say that high quality careers-related materials are in demand by schools now more than ever.

A key problem in STEM participation is the perception that studying science is only for those that aspire to become scientists (Archer and DeWitt, 2017). This is in contrast with the wide range of careers both related to and beyond science that further STEM education can enable. Therefore, this "science = scientists" link needs to be broken by highlighting to young people and their key influencers (e.g. teachers, parents/carers, community leaders) the prevalence and relevance of STEM subjects to everyday life and a diverse selection of potential career paths (Archer and DeWitt, 2017; Davenport et al., 2020; Archer et al., 2021).

Good practice towards diversity in communications more generally may be gleaned from the numerous efforts aimed at improving the diversity within STEM fields, due to the under-representation of women, disabled people, and those from ethnic-minorities or socially-disadvantaged groups (Campaign for Science and Engineering, 2014). One common approach is to strive for equal representation of minority demographics, in for instance role models, so that those aspiring towards STEM can see

"people like me" in those fields which may help tackle damaging stereotypes (e.g. Huntoon and Lane, 2007; Prinsley et al., 2016; González-Pérez et al., 2020). In other words, equal weight should be given to all categories, irrespective of whether they constitute a majority or minority within society.

In this paper, we investigate the representation of the space sector within current careers resources to ascertain whether they align with these educational recommendations.

## 30  2   Current space careers resources

The space sector involves a wide range of upstream (making and sending objects to space), downstream (using these objects to deliver products/services for exploitation), and ancillary (providing specialised support) roles, with scientific activities spanning all three (Sadlier et al., 2019; Sant et al., 2021; know.space, 2021). In the UK alone there are 45,100 space-related roles in industry (0.14% of the workforce), which support 126,300 jobs across the supply chain and generate £6.6 billion (0.30% of the 35  gross domestic product). The UK space sector is currently undergoing rapid development with many emerging opportunities (e.g. spaceports) that aim to drive further economic benefits. However, this can only be realised if there is a workforce trained and willing to undertake these new roles, highlighting the need for representation of space sector roles in careers education.

The 2020 Space Census was the first national survey of the UK space workforce (Thiemann and Dudley, 2021), inclusive of both industry and academia. It provides, to our knowledge, the best current classification scheme and breakdown of the 40  diverse roles present within the UK space sector. These are shown to the right of Figure 1a. This scheme and data are used as a benchmark for assessing current space careers resources.

We undertook desk research to find what careers resources for young people currently exist within the UK that aim to raise awareness and describe a range of roles across the space sector. Our search criteria meant that we could not include resources targeted at other countries (e.g. USA; Angeles and Vilorio, 2016), which promoted careers within specific organisations (e.g. 45  Serco, 2022), focused on only one aspect of the sector (e.g. Royal Academy of Engineering, 2018), simply listed current vacancies (e.g. Careersin.space, 2022), or just direct readers to other organisations (e.g. UK Space Agency, 2019). Only five sets with at least 9 roles (i.e. 1 per space census category if evenly distributed) were found: Edge Barrow School (2017), European Space Education Resource Office (2021), Royal Astronomical Society (2017), SpaceCareers.uk (2021), and University of Edinburgh Careers Service (2016). If other resources within our criteria exist, they likely do not have considerable reach or 50  impact. The jobs featured in each of the resources were classified using the space census scheme, performed independently by two people finding 91% agreement (Cohen's $\kappa = 0.9$, see Appendix A). Breakdowns of each set of resources by category are shown as the first four stacked plots in Figure 1a. We tested whether the resources either had an even split of categories, which would best reflect diversity, or were representative of the space census. This was done through chi-squared statistical tests (see Appendix A), with the outcomes listed in Figure 1b below each stacked plot.

Our results show that, to high confidence, none of the current space careers resources have a near-even split of 11% per category (corresponding to 1–3 roles). Indeed, between 2–4 of the categories are missing in each resource. Combining these

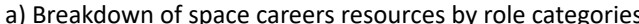

**a) Breakdown of space careers resources by role categories**

Legend: Other, Administration, Policy, Education, Sales, Computing, Management, Engineering, Scientific

Categories by resource:

- Edge Barrow School (n=18): Scientific 33%, Engineering 11%, Computing 11%, Sales 6%, Education 6%, Policy 6%, Other 28%
- ESERO (n=16): Scientific 44%, Engineering 13%, Management 19%, Computing 6%, Sales 6%, Other 13%
- RAS (n=15): Scientific 40%, Engineering 7%, Management 7%, Computing 13%, Education 7%, Policy 20%, Other 7%
- SpaceCareers.uk (n=29): Scientific 41%, Engineering 10%, Management 7%, Computing 7%, Policy 7%, Administration 3%, Other 24%
- University of Edinburgh (n=14): Scientific 50%, Engineering 21%, Sales 7%, Policy 7%, Other 14%
- This Work (n=36): Scientific 14%, Engineering 8%, Management 11%, Computing 8%, Sales 8%, Education 8%, Policy 6%, Administration 17%, Other 19%
- Space Census 2020 (n=1540): Scientific 27%, Engineering 26%, Management 18%, Computing 7%, Sales 6%, Education 5%, Policy 3%, Administration 2%, Other 5%

**b) Statistical tests on space career resources by role categories**

Hypothesis: Categories are uniformly distributed

| | Edge Barrow School | ESERO | RAS | SpaceCareers.uk | University of Edinburgh | This Work |
|---|---|---|---|---|---|---|
| $\chi^2(df=8)$ | 18.000 | 22.250 | 16.800 | 37.724 | 27.143 | *5.500* |
| p | 0.0233 | 0.0065 | 0.0346 | <0.0001 | 0.0002 | *0.7419* |

$\chi^2(df=10) = 59.773$ , $p = 4 \times 10^{-9}$

Hypothesis: Classifications are distributed as per Space Census 2020

| | Edge Barrow School | ESERO | RAS | SpaceCareers.uk | University of Edinburgh | This Work |
|---|---|---|---|---|---|---|
| $\chi^2(df=8)$ | 22.791 | *5.907* | 20.773 | 31.038 | *10.288* | 69.604 |
| p | 0.0132 | *0.6341* | 0.0166 | 0.0020 | *0.2247* | <0.0001 |

$\chi^2(df=10) = 33.178$ , $p = 3 \times 10^{-4}$

**Significantly different (reject hypothesis at α=0.05 level)** , *Consistent (do not reject hypothesis at α=0.05 level)*

**c) Roles in new space careers resource**

HUMAN RESOURCES, POLICY MAKER, BUSINESS DEVELOPMENT, STRUCTURAL/MECHANICAL ENGINEER, ARTIST, PROJECT MANAGER, SPACE PSYCHOLOGIST, SPACE NUTRITIONIST, ASTROBIOLOGIST, SPACE OPERATIONS NURSE, SCIENCE COMMUNICATOR, EARTH OBSERVATION, FINANCE, RISK MANAGEMENT, SPACE TRAVEL AGENT, TECHNICAL RECRUITER, DATA SCIENTIST, PRODUCT ASSURANCE, WEATHER, DATA ANALYST, TEACHING FELLOW, GROUND SOFTWARE, ASTROPHYSICIST, MUSEUM CURATOR, SPACE COMMAND, SUPPLY CHAIN, FLIGHT SURGEON, GEOLOGIST, JOURNALIST, ENVIRONMENTAL ENGINEER, SATELLITE SALES, INNOVATION MANAGER, COMMUNICATIONS, SYSTEMS ENGINEER, SPACE LAWYER, EXECUTIVE, FLIGHT SOFTWARE, INDEPENDENT COST ESTIMATOR

**Figure 1.** a) Breakdown of current UK space sector careers resources compared to the UK Space Census 2020 (Thiemann and Dudley, 2021). b) Outcomes of chi-squared statistical tests from these distributions. c) Word cloud of the space roles chosen for a new resource, where colours relate to the categories in panel a (font sizes have no meaning).

tests into an overall result (see Appendix A) shows this conclusion is highly robust. Therefore, current resources are perhaps not best representing the diversity of space-related careers available.

Comparing the resources to the space census, we find that all of them over-represent scientific careers. Given the low levels of young people aspiring towards being a scientist from an early age (Archer and DeWitt, 2017), it appears that these resources may perpetuate misconceptions about the usefulness of science. On the other hand, the large proportions of "Other" careers across most sets means that several less traditional career options related to space are being highlighted, which is advantageous. The statistics indicate the Edge Barrow School, RAS, and SpaceCareers.uk resources are highly unrepresentative of the UK space sector. We cannot confidently claim this for the others, though they have relatively small numbers of roles. Nonetheless, combining the results again yields a highly significant conclusion that current space careers resources are generally unrepresentative of the sector.

Finally, we note that these resources tend to be targeted at upper-secondary and university students. Therefore, there appears to be a lack of space-related careers material aimed at the ages most in need of engagement, i.e. primary and lower-secondary students (Archer et al., 2013; Holman, 2014; Davenport et al., 2020).

## 3 Developing a new resource

Given these findings, we endeavoured to create a more diverse set of UK-based space careers for a new resource to be aimed at younger ages. This was achieved by contacting Imperial Space Lab's industrial partners, reading reports on the UK space sector (e.g. Sadlier et al., 2019; Sant et al., 2021; know.space, 2021), finding advertised vacancies, and more general online research. The list of roles was iterated several times until it was felt the final set of 36 careers displayed in Figure 1c, greater in number than current resources, well captured the diversity of the sector.

Our aim was that this set would have near-equal numbers in each job category. The breakdown is shown as the fifth stacked plot in Figure 1a along with results of the statistical tests (panel b). These reveal that our set is indeed consistent with this aim, hence better represents the sector's diversity. Consequently, it is significantly different from the space census, though importantly no majority category from the census is over-represented. As with existing resources, "Other" careers form a significant fraction of the set thereby highlighting less traditional paths. It is also worth noting that the high number of roles in administration, i.e. relating to the running of a business or organisation, was deliberate since "business" is by far the most popular aspiration amongst young people (Archer et al., 2013). The application of these compiled roles into the design of a new careers resource is beyond the scope of this paper.

## 4 Conclusions

Educational research has revealed improved careers education, particularly for younger ages, may be required to improve participation in STEM. This needs to highlight the diversity of career options STEM subjects can enable, breaking the misconception that science is only for scientists. Focusing on space-related careers, we have found that currently available UK

resources perhaps do not best represent the diversity of roles present in the sector. In particular, there is an over-representation of scientists within them, which may perpetuate stereotypes. We have, therefore, compiled a more diverse set of space-related
careers which does not appear to suffer from these issues. These roles will form the basis of a new space careers resource for primary and lower-secondary students, which we hope will better align with the recommendations from recent educational research.

## Appendix A: Statistical methods

Cohen's $\kappa$ is a measure of reliability for coding categorical items (McHugh, 2012). It is calculated as

$$\kappa = \frac{p_0 - p_e}{1 - p_0}$$

where $p_0$ is the proportional agreement among coders and $p_e$ is that expected by chance. $\kappa$ ranges between 0 (consistent with random) and 1 (perfect agreement).

A chi-squared test compares observed frequencies $O_i$ within $k$ categories to those expected $E_i$ under some (null) hypothesis. The statistic is given by

100 $$\chi^2 \left(\mathrm{df} = k - 1\right) = \sum_{i=1}^{k} \frac{(O_i - E_i)^2}{E_i}$$

where $\mathrm{df}$ are the degrees of freedom. Due to small numbers, $p$-values (the probability of obtaining test results at least as extreme as those observed) were computed via 10,000 Monte Carlo simulations of $\chi^2$ for each resource's size under the hypotheses. If $p < 0.05$ then the observations are considered significantly different.

$p$-values of $n$ independent tests for the same hypothesis can be combined using Fisher's (1925) method to arrive at an overall
chi-squared statistic

$$\chi^2 \left(\mathrm{df} = 2n\right) = -2 \sum_{i=1}^{n} \ln \left(p_i\right)$$

whose $p$-value can be calculated.

*Data availability.* Data supporting the findings are derived from listed public domain resources.

*Author contributions.* MOA was involved in the conceptualization, funding acquisition, supervision, formal analysis, visualization, and
110 writing of this work. CLW and SD designed the methodology, performed the investigation, and undertook data curation. SF contributed to project administration and supervision. AP provided resources and assisted with validation.

*Competing interests.* The authors declare that they have no conflict of interest.

*Acknowledgements.* We thank the Jonathan Eastwood for his support in this project and research. This work has been made possible by Imperial College London's Undergraduate Research Opportunity Programme, through funding from the Department of Physics and Space Lab. M.O. Archer holds a UKRI (STFC / EPSRC) Stephen Hawking Fellowship EP/T01735X/1. We are also thankful for funding from STFC (ST/W00545X/1).

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
