# Peer review of "GC Insights: Space sector careers resources in the UK need a greater diversity of roles"

_Geoscience Communication, 2022_

## Author Response (AR1)

**Response to reviewers**

GC Insights: Space sector careers resources need a greater diversity of roles
M. O. Archer et al.

We thank both reviewers for their considered comments on our manuscript. Here we outline how we have taken them into account while revising the manuscript. All line numbers correspond to the tracked changes version of the revision.

**Anonymous Referee #1**

The paper is interesting in its concept but perhaps does not fully investigate the issues arising. There certainly are shortcomings in the diversity of roles demonstrated within careers resources, but work has been done in the last 5 years in order to address these roles, and this work has not been mentioned in this paper.

The paper is in the GC Insights format. As described by the journal's website "The format is meant for presenting a single insight that is likely to initiate further research or communication activity within the interdisciplinary readership of GC." This may play a part of the reviewer's comment about not fully exploring the issues arising, as these will form the subject of future work, both in engagement practice and educational research. Furthermore, GC Insights have a limit of 1,500 words, which the manuscript is already approaching. We are aware of recent research on placing people behind jobs with regards to careers engagement. However, the benefits of exposing young people to role models is complex and not possible to discuss within 1,500 words. The focus of this paper is on the awareness of the breadth of the space sector in careers information and we reference only work of direct relevance to this.

The paper does address relevant scientific questions within the scope of GC, and the analysis of the materials uses a novel technique to glean the comparison data. However, further analysis using this same process should be done on more careers information before relevant conclusions can be made.

We undertook thorough research into the available careers resources for young people currently exist within the UK that aim to raise awareness and describe a range of roles across the space sector. This resulted in only the four sets of resources presented, with no others found which fit our specific search criteria (please see our later comment about the RAS resources). Our search criteria meant that we could not include resources aimed at other countries (e.g. USA, https://www.bls.gov/careeroutlook/2016/article/careers-in-space.htm), which promoted careers within specific organisations (e.g. https://www.serco.com/eu/careers/careers-in-space), focused on only one aspect of the sector (e.g. https://www.thisisengineering.org.uk/what-interests-you/space/), simply listed current vacancies (e.g. https://www.careersin.space/), or signposted organisations' resources (e.g. https://ukspace.org/wp-content/uploads/2019/06/Careers-in-Space_UKSA.pdf). While this is appropriate given the GC Insights format, we have added a note about the types of resources we could not include in our study on lines 46-49. Nonetheless, despite the small number of sets of resources available to young people about space careers, they all show a clear and statistically significant trend, which is what we report on in this manuscript.

Whilst the analytical methods are innovative and appropriately mathematical, only applying them to four cases does not give sufficient enough data to fully support the interpretations and conclusions.

These may be only four sets of resources, but they cover many careers (77 in total). Many of the chi-squared tests applied to each resource were statistically significant. Combining the four independent statistical tests under the same null hypothesis performed using Fisher's method, we can arrive at overall results for space careers resources. These yield p-values of <0.0001 for both the uniform and space census hypotheses. Therefore, despite there being only four sets presented, the results are highly statistically robust. These tests have been added to Figure 1. Furthermore, the sets presented were the only relevant resources for young people about space careers found in the UK as previously mentioned. Indeed, several of them come from large organisations whose goals are to promote the space industry. Arguably two of the resources we found (Edge Barrow School and University of Edinburgh) are rather obscure. Therefore, if other resources within our criteria do exist, they likely do not have sufficient reach or impact as they have proven difficult to access (though again please see our later comment about the RAS resources). Hence, within the limits of this study, we are confident that sufficient data is presented to support our interpretation and conclusions.

Authors do give credit to related work and indicate their original contribution. The title does reflect the contents of the paper. The abstract does provide a concise summary. The overall presentation is clear and appropriately structured. The figure is clear and meaningful, although in order to understand all the data you do have to ensure you have understood the statistical mathematics that has been carried out – this might be a barrier to some readers.

We have revisited how the statistical methods are presented in the text, using an appendix for more mathematical detail so the main text can simply summarise the results for a broader audience.

The language is fluent and precise, but might be too analytical to give overall understanding for those who have not encountered these practices previously.

We have attempted to make the language more accessible for a broader audience.

I feel that there are too many references for a relatively short paper, although many quoted are excellent references.

The number of references is in line with published GC Insights papers within the journal, e.g. Hillier et al. (2022, https://doi.org/10.5194/gc-5-11-2022).

The conclusions of the paper describe 36 careers that must be in a careers resource in order for it to capture the diversity of the sector. These are valuable to note, but I worry that the authors would not be able to find specific examples of people in these careers, certainly not within the UK. Any careers resource that included these 36 career might be diverse, but may lose value if it is too long, too specific, or describing careers that are very obscure or unattainable.

We feel that the reviewer has made assumptions about the method in which our proposed careers resource would be implemented. This is outside of the scope of this paper and firmly within the future engagement practice and educational research motivated by this work that we will be undertaking in the future – in line with the purpose of the GC Insights format, as previously mentioned.

We do not present these careers as being necessary for a careers resource to be diverse – they are simply a list compiled that we show cover the space sector well. Most of the careers were decided upon from looking at current UK companies and their job listings, or contact with them directly and the materials they provided to us. Therefore, we feel it is incorrect to say that we wouldn't find examples of these roles in the UK. With regards to the particular choice of careers listed, the intention when choosing them was to avoid unattainable positions, such as "Astronaut" which are

often cited as a space career in existing resources. We don't believe that any of the listed roles are obscure. Potentially only "Space Travel Agent" might be classed as such, but we've seen space tourism grow in the past year alone and private space firms like Virgin Galactic are hiring a multitude of roles within the UK. Thus it isn't unreasonable to expect this growth to continue and further positions open by the time current upper-primary / lower-secondary pupils start a career.

We would like to point out that 36 careers is only slightly more than the 29 listed on SpaceCareers.uk. Furthermore, our intended format for the future careers resource is in attributes-based postcards, a short-form format designed to accommodate the number of roles presented in a suitable way. Therefore, we would like to reassure the reviewer that we have been considering many of their points in the preliminary development of our proposed space sector careers resource. However, the design of this proposed resource is beyond the scope of this GC Insights paper.

Analysis needs to be carried out on more careers resources, such as the careers booklet created by the Royal Astronomical Society: https://ras.ac.uk/education-and-careers/careers-booklet-sky-high-and-down-earth and other examples on the RAS website: https://ras.ac.uk/education-and-careers/careers

We thank the reviewer for these suggestions. The RAS careers booklet was not originally included in the study since it is slightly different to a dedicated space careers resource – as noted in its description the booklet highlights opportunities in *and away* from the space industry that studying astronomy and geophysics can enable. However, we now included it in the revised manuscript. Analysis shows the case studies presented within this resource follow very similar trends to the others already presented and the results of the statistical tests show the job categories covered in the case studies are inconsistent with both a uniform distribution ($p=0.0346$) and the space census ($p=0.0166$). Therefore, adding this does not change our conclusions. The other jobs highlighted on the RAS website are predominantly profiles of academics, with a few examples in areas such as teaching, outreach etc. Including these as a separate resource would show even stronger biases to the overrepresentation of scientific careers, hence would further support our conclusions. However, since they do not appear to be a curated set (unlike the booklet) we feel it appropriate to not include the website examples.

**Anonymous Referee #2**

The paper is relevant, interesting and well built. I think that addressing the comments by R1 will improve the quality of the paper, therefore I will not address issues that have already been raised. My review will for this reason only consist of a few minor comments.

We thank the reviewer for their positive comments on the paper.

It is worth noting that the space sector is developing quickly in the UK (see efforts for building spaceports in Scotland and Cornwall) which will drive economic development and create more roles. The economic development can only be fully realised if there is workforce who is trained and willing to undertake those roles (most of which, as the authors note are not currently accurately represented. In this view (appreciating that there is a word limit that must be respected) I suggest the paper would benefit of including a consideration on the fact that a rich representation of space careers will contribute to addressing the UK skills gap. I would also suggest to ask professionals working on the development of new space facilities what they think of the career range proposed and if they have suggestions on other future career paths in the UK that the authors may not have already considered or included.

We thank the reviewer for these suggestions. Points on the current development of the space sector in the UK, and thus the need for careers resources in this area for the target age group, have been added to the paper on lines 37-40. As to contact with professionals, we noted already in the paper that we discussed space sector roles with many industrial contacts in the development of the list presented, which included e.g. Spaceport Cornwall.

Another point that validates the need for good space career resources, is that often the existing ones consist of simply signposting to other organisations, instead of actually describing space careers in an age appropriate way for school pupils below A-level (see eg this resource from UKSA https://ukspace.org/wp-content/uploads/2019/06/Careers-in-Space_UKSA.pdf). A clearer explanation of what for the authors constitutes a "space careers resource" would help the reader appreciate this more.

In our research, careers resources are those which aim to raise awareness and describe a range of roles across the space sector, a definition we have added to the paper on lines 45-46. We found several cases like the reviewer raises, including that specific example, which simply point to the resources analysed in this paper, which we have now raised (line 49).

Though not freely available, S Kanani's book "How to be an Astronaut and Other Space Jobs" is a resource that has the target audience identified by this study (young people that are not yet about to decide their future career); it would be interesting to assess it according to the same criteria the authors used with the caveat that it is a book to be purchased, or made available by school libraries/local libraries. Given the current paucity of space career resources with diverse role models, Kanani's book would be appropriate to mention in this paper.

We thank the reviewer for this suggestion. While relevant to the broader topics touched upon by the paper, since as the reviewer notes this is a book to be purchased and not a freely available and dedicated space careers resource in the traditional sense, we feel it should not be included. Issues surrounding role models within the space sector is a far larger topic than can be contained within the word limit, therefore, we feel these aspects would better be discussed in the future work motivated by this paper rather than within this short paper itself.

My last comment concerns one of the careers that would consitute a more representative "career pack". The authors pick "Communications" as one of their 36 categories, but as someone with a background in the space sector, I am not entirely sure of what the word aims to describe - is it telecommunications engineering (eg ground segment to satellite communications, but this is maybe covered by the Ground Software role?) or a marketing oriented role? I suggest amending the name would clarify this ambiguity to a reader that hasn't seen the full set of career cards and provide an accurate representation of the language used in the sector.

We apologise for the confusion. This role was meant to cover the various internal and external communications roles, such as writing statements and press releases, working with the media, or liaising between different departments. We have changed it to Communications Executive, which hopefully will make the distinction between telecommunications as the reviewer raises. We have also included all the roles in the figure as a word cloud, coloured by our classifications, which should also help readers discern further meaning of the intended roles without having to see the full (in development) resource.

---

## Author Response (AR2)

**Response to editor**

GC Insights: Space sector careers resources in the UK need a greater diversity of roles
M. O. Archer et al.

We thank the editors for accepting our manuscript subject to some technical corrections. We outline here how we have responded to these.

1) I think the title of the paper should have "UK" in it (similar to the edit you correctly have made in the abstract). In addition to correctly representing the scope of your work, I think it will also be helpful for your article to appear in relevant search results.

Perhaps "GC Insights: Space sector career resources in the UK need a greater diversity of roles"

We have changed the title as the editor suggests.

2) In Fig 1c (the word cloud), do the different font sizes mean something? I suggest either making the font size equal or explaining its relevance in the caption.

The word cloud was produced by software that took the words along with their desired sizes and colours and fit these into a shape. Despite all the sizes being set as equal, the result was some words had different sizes. As were unable to find other software that would produce the desired outcome, we have simply added a note to the caption stating that the font sizes have no meaning.